# Intraoperative blood loss may be associated with myocardial injury after non-cardiac surgery

Jungchan Park[1☯], Ji-hye Kwon[1☯], Seung-Hwa Lee[2]*, Jong Hwan Lee[1], Jeong Jin Min[1], Jihoon Kim[2], Ah Ran Oh[1], Wonho Seo[1], Cheol Won Hyeon[2], Kwangmo Yang[3], Jin-ho Choi[2,4], Sang-Chol Lee[2], Kyunga Kim[5,6], Joonghyun Ahn[5], Hyeon-Cheol Gwon[2]

1 Department of Anesthesiology and Pain Medicine, Samsung Medical Center, Sungkyunkwan University School of Medicine, Seoul, Korea, 2 Division of Cardiology, Department of Medicine, Heart Vascular Stroke Institute, Samsung Medical Center, Sungkyunkwan University School of Medicine, Seoul, Korea, 3 Center for Health Promotion, Samsung Medical Center, Sungkyunkwan University School of Medicine, Seoul, Korea, 4 Department of Emergency Medicine, Samsung Medical Center, Sungkyunkwan University School of Medicine, Seoul, Republic of Korea, 5 Statistics and Data Center, Research Institute for Future Medicine, Samsung Medical Center, Seoul, Korea, 6 Department of Digital Health, SAIHST, Sungkyunkwan University, Seoul, Korea

☯ These authors contributed equally to this work.
* shuaaa.lee@samsung.com

**Data Availability Statement:** The data we used for this study was curated using Clinical Data Warehouse (CDW) which psuedonomynize the data from our institutional electronic medical records. So, our data is deidentified by eliminating

## Abstract

### Background

This study aimed to evaluate the association between intraoperative blood loss and myocardial injury after non-cardiac surgery (MINS), which is a severe and common postoperative complication.

### Methods

We compared the incidence of MINS based on significant intraoperative bleeding, defined as an absolute hemoglobin level < 7 g/dL, a relative hemoglobin level less than 50% of the preoperative measurement, or need for packed red cell transfusion. We also estimated a threshold for intraoperative hemoglobin level associated with MINS.

### Results

We stratified a total of 15,926 non-cardiac surgical patients with intraoperative hemoglobin and postoperative cardiac troponin (cTn) measurements according to the occurrence of significant intraoperative bleeding; 13,416 (84.2%) had no significant bleeding while 2,510 (15.8%) did have significant bleeding. After an adjustment with inverse probability weighting, the incidence of MINS was higher in the significant bleeding group (35.2% vs. 16.4%; odds ratio, 1.58; 95% confidence interval, 1.43–1.75; p < 0.001). The threshold of intraoperative hemoglobin associated with MINS was estimated to be 9.9 g/dL with an area under the curve of 0.643.

all identifiable variables such as name, social security number, hospital number, and etc. However, it is illegal to open this data to the public without restriction. Regarding the availability of our data, please contact jong-hwan.park@samsung.com, the head of our institutional data security department.

**Funding:** Unfunded studies.

**Competing interests:** No authors have competing interests.

## Conclusion

Intraoperative blood loss appeared to be associated with MINS. Further studies are needed to confirm these findings.

## Clinical registration

The cohort was registered before patient enrollment at https://cris.nih.go.kr (KCT0004244).

## Introduction

Myocardial injury after non-cardiac surgery (MINS) has recently arisen as a strong predictor of postoperative mortality [1]. Occurring in an estimated 8 million cases out of more than 200 million non-cardiac surgery patients every year, MINS has become the most common postoperative state related to mortality [2]. Massive blood loss is frequently encountered during surgical procedures and can lead to decreased hemoglobin levels. Oxygen supply/demand mismatch is a proposed as one of the mechanism for MINS [3]. Therefore, the compromised balance between oxygen supply/demand due to decreased hemoglobin levels and oxygen content is likely to be related to MINS [4], but the association between intraoperative blood loss and the occurrence of MINS has never been investigated.

Intraoperative blood loss and decreased hemoglobin levels are independently associated with perioperative mortality and serious morbidity such as myocardial infarction, stroke, and renal failure [5–7]. Furthermore, blood replacement by packed red blood cell (RBC) transfusion may lead to other severe side effects, complicating the perioperative clinical situation when the risks of anemia and transfusion needs to be balanced [4, 6, 8, 9]. Indeed, increased need for transfusion is one of the mechanisms of perioperative morbidity in anemic patients, and it is difficult to specify whether the morbidity originates from anemia or transfusion. Therefore, in this study, we evaluated whether intraoperative blood loss is associated with MINS. We also separately selected patients with actual hemoglobin decreases and those who required transfusion of packed RBCs using a de-identified real-world cohort.

## Materials and methods

Because this registry was generated in a de-identified format, the Institutional Review Board at Samsung Medical Center waived the need for approval for this study and the requirement for written informed consent for access to the registry (SMC 2019-08-048). The cohort was registered before patient enrollment at https://cris.nih.go.kr (KCT0004244).

### Study population and data collection

Our institution operates a paperless electronic medical record system that contains data from more than 4 million patients with more than 2 million surgeries, 900 million laboratory findings, and 200 million prescriptions. We developed "Clinical Data Warehouse Darwin-C," an electronic system designed for investigators to search and retrieve de-identified medical records from our electronic archive system. Using these systems, we generated the SMC-TINCO registry (Samsung Medical Center Troponin in Non-cardiac Operation), a large single-center cohort containing data from 43,019 consecutive patients who had cTn I levels measured before or within 30 days after non-cardiac surgery between January 2010 and June 2019.

Our exclusion criteria for this study were: 1) patients who were younger than 18 at the time of surgery, 2) patients without postoperative cTn I measurements, 3) patients who had cardiac massage before the diagnosis of MINS, and 4) patients without intraoperative hemoglobin level measurements. Patients were divided according to significant intraoperative bleeding. Significant intraoperative bleeding was defined as an intraoperative lowest hemoglobin level < 7 g/dL, an intraoperative lowest hemoglobin level less than 50% of the preoperative measurement, or need for intraoperative packed RBC transfusion [10–13].

## Study outcomes and definitions

The primary outcome was MINS, which was defined as a peak cTn I level above the 99th percentile of the upper reference limit within 30 days after surgery. Elevations of cTn from non-ischemic etiology such as sepsis, pulmonary embolus, atrial fibrillation, cardioversion, or chronic elevation were excluded [3, 14]. Among 3,193 patients with postoperative cTn elevation, 110 patients were diagnosed with a non-ischemic cause, and 3,083 (19.4%) patients were diagnosed with MINS. Preoperative anemia was defined as a last preoperative hemoglobin level < 13 g/dL in men and < 12 g/dL in women [15]. Active cancer was defined as a histologic diagnosis of cancer within the 6 months prior to surgery [16]. High-risk surgery was defined as procedures with reported mortality risk >5% according to the 2014 European Society of Cardiology/Anesthesiology guidelines [17].

## Perioperative cTn I measurement and blood management

Perioperative cTn I was measured for patients with moderate or high cardiovascular risk based on current guidelines [17], but it was also measured in patients with mild risk at the discretion or request of an attending clinician. An automated analyzer (Advia Centaur XP, Siemens Healthcare Diagnostics, Erlangen, Germany) was used. The lowest limit of detection was 6 ng/L, and the 99th percentile upper reference limit provided by the manufacturer was 40 ng/L [18]. Patients with elevated cTn levels were referred to cardiologists for further evaluation and management.

Preoperative hemoglobin measurement was included in routine preoperative blood tests, and packed RBCs were preoperatively transfused in anemic patients at the discretion of an attending clinician or at the request of an anesthesiologist. During the intraoperative period, hemoglobin measurement was not a routine procedure, but was selectively performed in patients suspected to have massive bleeding. Our indication for intraoperative transfusion of packed RBCs changed during the study period. Following the current guidelines at the time, the threshold of hemoglobin level for RBC transfusion was changed from 10 g/dL to 7 g/dL in January 2017 [10–12]. Intraoperative transfusion was also performed at the discretion of the attending anesthesiologist or surgeon in patients with higher cardiovascular risk or anticipated massive bleeding.

## Statistical analysis

For continuous variables, differences were compared by the t-test or the Mann-Whitney test and presented as the mean ± standard deviation or median with interquartile range. Categorical data are presented as number (%) and compared by using the chi-square or Fisher's exact test, as applicable. The incidence of MINS was compared using a stratified logistic regression model and was reported as an adjusted odds ratio (OR) with 95% confidence interval (CI). Mortalities were compared with Cox regression analysis and reported as the hazard ratio (HR) with 95% CI. To retain a large sample size and maximize the study power while maintaining a balance in covariates between the two groups, we conducted rigorous adjustment for

differences in baseline patient characteristics using weighted regression models with inverse probability weighting (IPW) [19]. According to this technique, weights for patients without significant bleeding were the inverse of the propensity score and weights for patients with significant bleeding were the inverse of 1 –the propensity score. For sensitivity analysis, we evaluated whether the observed association was significant in patients with chronic kidney disease and heart failure; we also tested whether this was significant after January 2017, when the threshold of hemoglobin for intraoperative transfusion was lowered to 7 g/dL. We also estimated the potential impact of unmeasured confounders by calculating the change in OR and CI according to associations of unmeasured confounders with significant bleeding and MINS, using an assumed unmeasured confounder with a prevalence of 40% [20]. To assess the efficacy of intraoperative hemoglobin level in predicting MINS, Pearson's correlation coefficient and receiver-operating characteristic (ROC) plots were constructed to estimate the threshold for intraoperative hemoglobin level and compute the specificity and sensitivity. Statistical analyses were performed with R 3.6.1 (Vienna, Austria; http://www.R-project.org/). All tests were 2-tailed and $p < 0.05$ was considered statistically significant.

## Results

Out of 43,019 patients in the registry, we excluded 1,154 patients who were younger than 18 at the time of surgery, 6,596 patients without postoperative cTn I measurements, 46 patients who had cardiac massage before the diagnosis of MINS, and 19,297 patients who did not have intraoperative hemoglobin levels. We included 15,926 patients in the final analysis (Fig 1).

Patients were divided into two groups based on the presence of significant intraoperative bleeding: 13,416 (84.2%) patients had no significant bleeding, while 2,510 (15.8%) patients did have significant intraoperative bleeding (Table 1). After adjustment with multivariable analysis, the risk for MINS was significantly increased in the significant bleeding group (35.2% vs. 16.4%; OR, 1.80; 95% CI, 1.61–2.00; $p < 0.001$) (Table 2). After IPW adjustment, there was still an increased risk for MINS in the significant bleeding group (OR, 1.58; 95% CI, 1.43–1.75; $p < 0.001$) (Table 2). Subgroup analysis demonstrated that the association between significant bleeding and MINS was confounded by emergency surgery. The risk of MINS was significantly increased in patients with significant bleeding when undergoing elective surgery, but not when undergoing emergency surgery (Fig 2). In our sensitivity analysis, this association was consistently significant in patients with chronic kidney disease and heart failure, and after the hemoglobin threshold for intraoperative transfusion was lowered in 2017 (S1 Table). Significant bleeding was associated with MINS, even if the assumed unmeasured confounders were present in one of our groups with a prevalence up to 40%. (S2 Table).

Patients were also stratified into four groups to account for both hemoglobin decrease and the use of intraoperative RBC transfusion. There were 13,416 (84.2%) patients who had no hemoglobin decrease and no transfusion, 1,937 (12.2%) who had no hemoglobin decrease with transfusion, 275 (1.7%) who had a hemoglobin decrease without transfusion, and 298

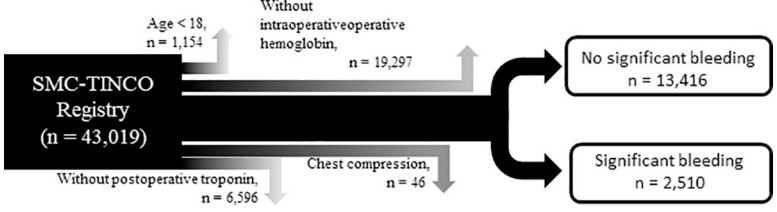

**Fig 1. Patient flowchart.**

**Table 1. Baseline characteristics according to significant intraoperative bleeding.**

| | Entire population | | | IPW | | |
|---|---|---|---|---|---|---|
| | No significant bleeding (N = 13416) | Significant bleeding (N = 2510) | ASD | No significant bleeding (N = 13370.5) | Significant bleeding (N = 2644.5) | ASD |
| Male sex | 8133 (60.6) | 1562 (62.2) | 3.3 | 8145.0 (60.9) | 1658.9 (62.7) | 3.7 |
| Age (years) | 61.7 (±13.5) | 60.0 (±13.5) | 12.4 | 61.5 (±13.6) | 62.2 (±13.3) | 5.0 |
| Preoperative anemia | 4949 (36.9) | 1646 (65.6) | 59.9 | 5497.0 (41.1) | 1064.8 (40.3) | 1.7 |
| Diabetes | 7890 (58.8) | 1748 (69.6) | 22.7 | 8019.3 (60.0) | 1446.4 (54.7) | 10.7 |
| Hypertension | 7290 (54.3) | 1424 (56.7) | 4.8 | 7331.9 (54.8) | 1436.0 (54.3) | 1.1 |
| Current smoking | 1362 (10.2) | 236 (9.4) | 2.5 | 1337.6 (10.0) | 282.8 (10.7) | 2.3 |
| Current alcohol | 2798 (20.9) | 360 (14.3) | 17.2 | 2639.6 (19.7) | 556.3 (21.0) | 3.2 |
| Chronic kidney disease | 680 (5.1) | 332 (13.2) | 28.6 | 850.8 (6.4) | 166.3 (6.3) | 0.3 |
| History of ischemic heart disease | 1913 (14.3) | 365 (14.5) | 0.8 | 1939.1 (14.5) | 395.5 (15.0) | 1.3 |
| History of heart failure | 282 (2.1) | 45 (1.8) | 2.2 | 277.8 (2.1) | 51.3 (1.9) | 1.0 |
| History of stroke | 938 (7.0) | 161 (6.4) | 2.3 | 926.8 (6.9) | 176.3 (6.7) | 1.1 |
| History of arrhythmia | 882 (6.6) | 161 (6.4) | 0.6 | 875.4 (6.5) | 179.4 (6.8) | 0.9 |
| History of heart valve disease | 149 (1.1) | 21 (0.8) | 2.8 | 143.2 (1.1) | 25.9 (1.0) | 0.9 |
| Active cancer | 7422 (55.3) | 1112 (44.3) | 22.2 | 7216.2 (54.0) | 1555.1 (58.8) | 9.8 |
| Preoperative care | | | | | | |
| RBC transfusion | 608 (4.5) | 151 (6.0) | 6.6 | 638.0 (4.8) | 123.2 (4.7) | 0.5 |
| Intensive care unit | 424 (3.2) | 278 (11.1) | 31.2 | 574.7 (4.3) | 111.4 (4.2) | 0.4 |
| ECMO | 0 | 1 (0.0) | 2.8 | 0 | 0.2 (0.0) | 1.1 |
| Continuous renal replacement therapy | 18 (0.1) | 54 (2.2) | 19.1 | 54.7 (0.4) | 11.3 (0.4) | 0.3 |
| Ventilator | 67 (0.5) | 56 (2.2) | 15.0 | 101.2 (0.8) | 18.9 (0.7) | 0.5 |
| Operative variables | | | | | | |
| ESC/ESA surgical high risk | 3997 (29.8) | 1258 (50.1) | 42.4 | 4306.0 (32.2) | 748.8 (28.3) | 8.5 |
| Emergency operation | 1521 (11.3) | 549 (21.9) | 28.6 | 1745.6 (13.1) | 354.4 (13.4) | 1.0 |
| General anesthesia | 13305 (99.2) | 2500 (99.6) | 5.5 | 13267.8 (99.2) | 2610.5 (98.7) | 5.2 |
| Operation duration, hours | 3.94 (±2.14) | 5.57 (±2.86) | 64.4 | 4.20(±2.51 | 4.16±2.20) | 1.7 |
| Continuous infusion of inotropes | 4592 (34.2) | 1278 (50.9) | 34.2 | 4855.3 (36.3) | 798.6 (30.2) | 13.0 |
| Types of surgery | | | >0.99 | | | |
| Vascular | 981 (7.3) | 169 (6.7) | | | | |
| Orthopediatric | 667 (5.0) | 146 (5.8) | | | | |
| Neuro | 3317 (24.7) | 166 (6.6) | | | | |
| Breast or Endo | 148 (1.1) | 22 (0.9) | | | | |
| Plastic or otolaryngeal or eye | 279 (2.1) | 53 (2.1) | | | | |
| Transplantation | 300 (2.2) | 768 (30.6) | | | | |
| Gynecology or urology | 823 (6.1) | 258 (10.3) | | | | |
| Gastrointestinal | 4559 (34.0) | 741 (29.5) | | | | |
| Noncardiac thoracic | 2328 (17.4) | 184 (7.3) | | | | |
| Other | 14 (0.1) | 3 (0.1) | | | | |

Data are presented as n (%) or mean (±standard deviation).

IPW, inverse probability weighting; ASD, absolute standardized mean difference; RBC, red blood cell; ECMO, extracorporeal membranous oxygenation; RAAS, renin-angiotensin-aldosterone system; ESC, European Society of Cardiology; ESA, European Society of Anesthesiology.

(1.9%) who had hemoglobin decrease with transfusion (S3 Table). Compared to patients who had no hemoglobin decrease and no transfusion, the risk for MINS increased according to both hemoglobin decrease and receipt of RBC transfusion (OR, 2.04; 95% CI, 1.83–2.27;

**Table 2. The incidence of myocardial injury after noncardiac surgery and mortality.**

| | No significant bleeding (N = 13416) | Significant bleeding (N = 2510) | Univariable analysis | | Multivariable analysis | | IPW analysis | |
|---|---|---|---|---|---|---|---|---|
| | | | Unadjusted OR/HR (95% CI) | p value | Adjusted OR/HR (95% CI) | p value | Adjusted OR/HR (95% CI) | p value |
| MINS | 2200 (16.4) | 883 (35.2) | 2.77 (2.52–3.04) | < 0.001 | 1.80 (1.61–2.00) | < 0.001 | 1.58 (1.43–1.75) | < 0.001 |
| 30-day mortality | 173 (1.3) | 112 (4.5) | 3.52 (2.77–4.46) | < 0.001 | 2.04 (1.55–2.67) | < 0.001 | 2.51 (1.91–3.28) | < 0.001 |
| Cardiovascular | 50 (0.4) | 28 (1.1) | 3.04 (1.91–4.82) | < 0.001 | 1.92 (1.13–3.27) | 0.02 | 1.90 (1.10–3.29) | < 0.001 |
| Noncardiovascular | 123 (0.9) | 84 (3.3) | 3.71 (2.81–4.90) | < 0.001 | 2.10 (1.53–2.88) | < 0.001 | 2.76 (2.02–3.76) | < 0.001 |

Data are presented as n (%).

MINS was presented with OR, and mortalities were presented as HRs.

IPW, inverse probability weighting; MINS, myocardial injury after noncardiac surgery; OR, odds ratio; HR, hazard ratio; CI, confidence interval.

$p < 0.001$ for no hemoglobin decrease with transfusion; OR, 6.13; 95% CI, 4.81–7.82; $p < 0.001$ for hemoglobin decrease without transfusion; and OR, 8.66; 95% CI, 6.77–10.96; $p < 0.001$ for hemoglobin decrease with transfusion) (Table 3). When the patients were solely

| Subgroups | No significant bleeding | Significant bleeding | OR (95% CI) | P-value | P for interaction |
|---|---|---|---|---|---|
| No hypertension | 6126 | 1086 | 1.65 (1.41 - 1.93) | <0.001 | 0.047 |
| Hypertension | 7290 | 1424 | 1.34 (1.18 - 1.53) | <0.001 | |
| No diabetes | 12636 | 762 | 1.11 (0.95 - 1.30) | 0.180 | <0.001 |
| Diabetes | 7890 | 1748 | 1.76 (1.54 - 2.00) | <0.001 | |
| No coronary artery disease | 11503 | 2145 | 1.57 (1.41 - 1.76) | <0.001 | 0.001 |
| Coronary artery disease | 1913 | 365 | 1.01 (0.79 - 1.28) | 0.943 | |
| No preoperative anemia | 8467 | 864 | 1.40 (1.21 - 1.61) | <0.001 | 0.379 |
| Preoperative anemia | 4949 | 1646 | 1.53 (1.32 - 1.77) | <0.001 | |
| No intraoperative inotropics | 8824 | 1232 | 1.30 (1.13 - 1.49) | <0.001 | <0.001 |
| Intraoperative inotropics | 4592 | 1278 | 1.94 (1.66 - 2.27) | <0.001 | |
| No high-risk surgery | 9419 | 1252 | 1.20 (1.06 - 1.36) | 0.003 | <0.001 |
| High-risk surgery | 3997 | 1258 | 2.15 (1.81 - 2.56) | <0.001 | |
| No emergency operation | 11895 | 1991 | 1.61 (1.44 - 1.80) | <0.001 | <0.001 |
| Emergency operation | 1521 | 549 | 0.97 (0.76 - 1.22) | 0.771 | |
| No active cancer | 5994 | 1398 | 1.52 (1.31 - 1.75) | <0.001 | 0.616 |
| Active cancer | 7422 | 1112 | 1.44 (1.25 - 1.65) | <0.001 | |

**Fig 2. Forest plot for subgroup analysis.**

**Table 3. The incidence of myocardial after noncardiac surgery and mortality according to intraoperative hemoglobin decrease and transfusion.**

|  | No hemoglobin decrease without transfusion (N = 13416) | No hemoglobin decrease with transfusion (N = 1937) | Hemoglobin decrease without transfusion (N = 275) | Hemoglobin decrease with transfusion (N = 298) |
|---|---|---|---|---|
| MINS, No (%) | 2200 (16.4) | 549 (28.3) | 147 (53.5) | 187 (62.8) |
| Unadjusted OR (95% CI) | 1 [reference] | 2.02 (1.81–2.24) | 5.85 (4.60–7.46) | 8.59 (6.77–10.94) |
| *p* value |  | < 0.001 | < 0.001 | < 0.001 |
| 30-day mortality, No (%) | 173 (1.3) | 61 (3.1) | 26 (9.5) | 25 (8.4) |
| Unadjusted HR (95% CI) |  | 2.46 (1.84–3.30) | 7.75 (5.14–11.71) | 6.76 (4.45–10.29) |
| *p* value |  | < 0.001 | < 0.001 | < 0.001 |
| Cardiovascular mortality, No (%) | 50 (0.4) | 17 (0.9) | 9 (3.3) | 2 (0.7) |
| Unadjusted HR (95% CI) |  | 2.37 (1.37–4.11) | 9.23 (4.54–18.78) | 1.87 (0.45–7.67) |
| *p* value |  | < 0.001 | < 0.001 | 0.39 |
| Non-cardiovascular mortality, No (%) | 123 (0.9) | 44 (2.3) | 17 (6.2) | 23 (7.7) |
| Unadjusted HR (95% CI) |  | 2.50 (1.77–3.52) | 7.14 (4.30–11.86) | 8.76 (5.61–13.67) |
| *p* value |  | < 0.001 | < 0.001 | < 0.001 |

Data are presented as n (%).

MINS was presented with OR, and mortalities were presented as HRs.

MINS, myocardial injury after noncardiac surgery; OR, odds ratio; HR, hazard ratio; CI, confidence interval.

divided according to hemoglobin decrease without considering RBC transfusion, 15,353 (96.4%) patients had no hemoglobin decrease whereas 573 (3.6%) patients did have decreased hemoglobin (S4 Table). The incidence of MINS was substantially increased in the decreased hemoglobin group (58.3% vs. 17.9%; OR, 3.28; 95% CI, 2.70–4.00; *p* < 0.001) (S5 and S6 Tables).

The threshold for intraoperative hemoglobin level associated with MINS was estimated to be 9.9 g/dL in ROC analysis, and the area under the ROC curve was 0.643. Using this value, the sensitivity and specificity were 45.2 and 76.8%, respectively (Fig 3).

## Discussion

The main finding of this study was that the incidence of MINS was associated with intraoperative blood loss. The calculated threshold for intraoperative hemoglobin level associated with MINS was 9.9 g/dL. Our findings suggest that maintaining an adequate intraoperative hemoglobin level may be helpful for preventing MINS.

Intraoperative blood loss is a well-established risk factor for postoperative mortality and morbidity [6, 7]. Despite various possible causes, the most frequently proposed mechanism for MINS is oxygen supply/demand mismatch, and our previous study also showed that preoperative anemia was associated with MINS [21]. In this follow-up study, we demonstrated that blood loss during surgical procedures could also increase MINS. So, our results on the association between intraoperative hemoglobin decrease and the increased incidence of MINS can be explained by reduced oxygen-carrying capacity compromising the myocardial oxygen supply while simultaneously requiring higher cardiac output to maintain adequate systemic circulation [22]. Moreover, the cardiac effects of oxygen supply/demand mismatch can be aggravated in a high cardiac output state such as the perioperative period [14]. In this study, we

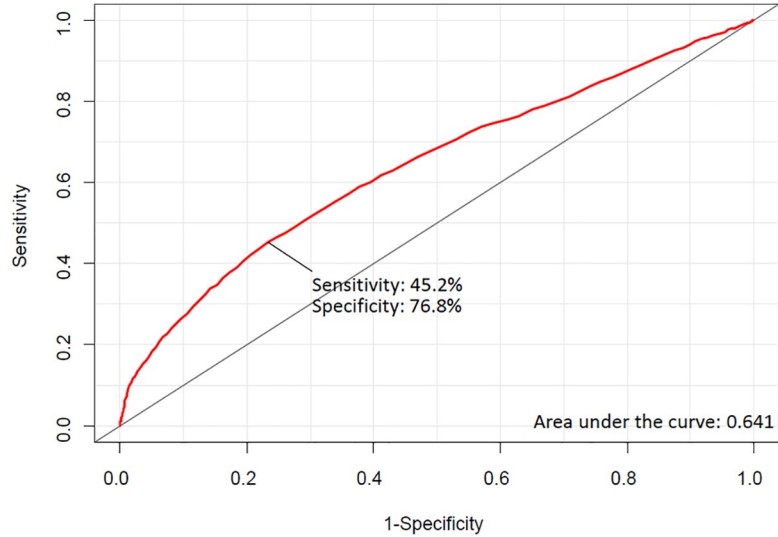

**Fig 3. Receiver-operating characteristic curves for intraoperative hemoglobin level associated with myocardial injury after non-cardiac surgery.**

demonstrated an association between blood loss and MINS, reinforcing the importance of avoiding intraoperative anemia. However, there are issues that require discussion.

First, definitive criteria for intraoperative blood loss have not been established, and the definitions of bleeding vary between studies on surgical patients. In this study, we selected patients whose intraoperative hemoglobin level decreased below 7 g/dL, which is generally assumed to be the level that most patients can tolerate [10–12, 23]. However, by selecting patients only according to absolute hemoglobin level, the amount of bleeding in patients with a high baseline hemoglobin may be underestimated, or the degree of intraoperative anemia that can be safely tolerated in chronically anemic patients may be overlooked. Therefore, we also applied a relative hemoglobin threshold of less than 50% from baseline to define significant bleeding, which has previously been associated with adverse outcomes [13]. Our primary analysis based on these composite criteria demonstrated a significant association between intraoperative bleeding and MINS. However, in this setting, it was uncertain whether the increased incidence of MINS was caused by the low hemoglobin level or receipt of transfusion, the side effects of which could also elevate cardiac troponin [24]. Therefore, we further conducted an analysis by separately considering RBC transfusion during the surgery and the actual decrease in hemoglobin.

The four-group comparison according to hemoglobin reduction and receipt of RBC transfusion showed that groups with decreased hemoglobin had increased MINS risk regardless of transfusion status. In addition, MINS risk was greater for patients with actual hemoglobin reduction regardless of transfusion status. Together, these results suggest that maintaining an adequate hemoglobin level may be more effective in preventing MINS than minimizing RBC transfusion. Previous studies also showed that intraoperatively transfused RBCs could not improve oxygenation, which is critical for MINS prevention [25, 26]. In addition, in patients with a hemoglobin decrease, the incidence of MINS was increased by RBC transfusion, suggesting that transfusion side effects might also contribute to the occurrence of MINS [4, 6, 9]. Despite the current guidelines advocating restrictive use of blood transfusion during surgical procedures, our findings suggest that the hemoglobin threshold for intraoperative transfusion may be higher than 7g/dL in order to prevent MINS [10–12, 17]. However, the net effect of intraoperative transfusion has long been controversial and inconclusive [8].

The observed association between intraoperative bleeding was significant in most of subgroups with risk factors for MINS. However, it was not significant in patients undergoing emergency surgery. This may be because patients requiring emergency surgery include those with preoperative hemodynamic instability or massive bleeding, which could be strongly associated with cTn elevation. These events causing a large fluctuation of hemoglobin levels from the preoperative period might have diluted the effect of intraoperative blood loss. In addition, the observed association between intraoperative blood loss and MINS was significant before and after lowering the institutional threshold for intraoperative RBC transfusion from 10 g/dL to 7 g/dL at January 2017.

The threshold of intraoperative hemoglobin level associated with MINS was estimated to be 9.9 g/dL in this study, higher than the currently suggested guidelines [10–12]. Instead, this estimated level is in line with the levels recommended for patients with high cardiovascular risks such as anemia, bleeding, and ischemic disease [10, 27]. Indeed, this study was conducted among 15,926 patients who needed intraoperative hemoglobin measurements; a larger number of patients (19,297) who did not need intraoperative hemoglobin measurement were excluded. Moreover, the entire registry inherently contains patients with some level of cardiovascular risk who were selected for perioperative cTn measurements. Another limitation of this estimate is that the area under the ROC curve was 0.643, reflecting a relatively low predictive value. This may be due to multifactorial nature of MINS and suggests that factors other than blood loss and receipt of transfusion need to be considered to predict MINS.

Our study has several limitations. First, this is a single-center, observational study; the possibility of selection bias or unmeasured confounding factors exists. In addition, our data may not be generalizable to populations in other countries considering ethnic differences in blood management. In addition, this study contains all types of noncardiac surgery, and the result might be different in particular types of surgery. Second, perioperative cTn I measurement was not included as a routine clinical practice at our institution. Although the measurement followed the institutional protocol, it was usually performed in patients with a high cardiovascular risk; therefore, our results may have been exaggerated by selection bias. Third, a detailed protocol for perioperative blood management was absent, and intraoperative hemoglobin measurement was selectively performed. In addition, RBC transfusion was treated as a binary variable in the analyses; the number of transfused packed RBC units was not considered. Despite these limitations, this is the first study to evaluate the occurrence of MINS according to intraoperative blood loss, suggesting the need for further studies on this subject.

## Conclusion

Intraoperative blood loss appeared to be associated with MINS, and further studies are needed.

## Supporting information

**S1 Table. Sensitivity analysis of the observed association between significant bleeding and myocardial injury after noncardiac surgery.**
(DOCX)

**S2 Table. Sensitivity analysis of the effect of an unmeasured confounder on odds ratio of significant bleeding for myocardial injury after noncardiac surgery.**
(DOCX)

**S3 Table. Baseline characteristics according to the actual hemoglobin decrease and intraoperative transfusion.**
(DOCX)

**S4 Table. Baseline characteristics according to the actual hemoglobin decrease without regarding intraoperative transfusion.**
(DOCX)

**S5 Table. The incidence of myocardial injury after noncardiac surgery and mortality according to the actual hemoglobin decrease without regarding intraoperative transfusion.**
(DOCX)

**S6 Table. Sensitivity analysis of the effect of an unmeasured confounder on odds ratio of hemoglobin decrease for myocardial injury after noncardiac surgery.**
(DOCX)

## Author Contributions

**Conceptualization:** Jungchan Park, Ji-hye Kwon, Seung-Hwa Lee.

**Data curation:** Jihoon Kim, Ah Ran Oh, Wonho Seo, Cheol Won Hyeon.

**Formal analysis:** Kyunga Kim, Joonghyun Ahn.

**Methodology:** Jungchan Park, Ji-hye Kwon, Seung-Hwa Lee, Kyunga Kim, Joonghyun Ahn.

**Supervision:** Jong Hwan Lee, Jeong Jin Min, Kwangmo Yang, Jin-ho Choi, Sang-Chol Lee, Hyeon-Cheol Gwon.

**Writing – original draft:** Jungchan Park, Ji-hye Kwon.

**Writing – review & editing:** Seung-Hwa Lee, Jong Hwan Lee, Jeong Jin Min, Hyeon-Cheol Gwon.

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
