## [Decision Letter · Decision Letter 0]

24 Jun 2020

PONE-D-20-14264

Intraoperative blood loss may be associated with myocardial injury after non-cardiac surgery

PLOS ONE

Dear Dr. Lee,

Thank you for submitting your manuscript to PLOS ONE. After careful consideration, we feel that it has merit but does not fully meet PLOS ONE’s publication criteria as it currently stands. Therefore, we invite you to submit a revised version of the manuscript that addresses the points raised during the review process.

Specifically, strongly consider the suggestions of reviewer one by minimizing exclusion criteria for the overall analysis and conduct sensitivity analyses based on the specific exclusions.

We look forward to receiving your revised manuscript.

Kind regards,

Wen-Chih Wu, MD, MPH

Academic Editor

PLOS ONE

Journal Requirements:

2. Please include captions for your Supporting Information files at the end of your manuscript, and update any in-text citations to match accordingly. Please see our Supporting Information guidelines for more information: http://journals.plos.org/plosone/s/supporting-information

Reviewers' comments:

Reviewer's Responses to Questions

**Comments to the Author**

1. Is the manuscript technically sound, and do the data support the conclusions?

Reviewer #1: Partly

Reviewer #2: Partly

2. Has the statistical analysis been performed appropriately and rigorously? 

Reviewer #1: Yes

Reviewer #2: I Don't Know

3. Have the authors made all data underlying the findings in their manuscript fully available?

Reviewer #1: No

Reviewer #2: Yes

4. Is the manuscript presented in an intelligible fashion and written in standard English?

Reviewer #1: Yes

Reviewer #2: Yes

5. Review Comments to the Author

Reviewer #1: I have uploaded an attachment as a word file xxxxxxxxxxxxxxxxxxxxxxxxxxxxxxxxxxxxxxxxxxxxxxxxxxxxxxxxxxxxxxxxxxxxxxxxxxxxxxxxxxxxxxxxxxxxxxxxxx

Reviewer #2: Overall well written paper confirming pre-existing literature.

Database/registry study with many limitations, most of which were addressed in the text of the paper. They acknowledged that more studies (more rigorously conducted studies to be specific) need to be conducted to support their conclusions and essentially this study is mostly hypothesis generating.

Large population included in study and overall well conducted (given limitations of registry) so does add some controversy to literature that is worth considering (changing transfusion 'triggers' in high risk populations to avoid MINS).

There were significant differences in baseline characteristics in patients who had blood loss versus those that didn't therefore there were many confounding factors that could influence reasons for these patients to have more MINS besides just blood loss, not sure all of this was accounted for by the statistics or fully explained/justified.

Few errors in supplemental tables that need to be corrected - Supplemental Table 1 has a category of surgery "Orthopediatric" since all patients were above age 18 I can only assume this should be 'Orthopedic'. Supplemental Table 3 says "ionotropics" should read 'ionotropes'.

6. PLOS authors have the option to publish the peer review history of their article (what does this mean?). If published, this will include your full peer review and any attached files.

Reviewer #1: No

Reviewer #2: No

---

## [Author Response · Author response to Decision Letter 0]

17 Jul 2020

July, 2020

Dr. Wen Chih-Wu 

Academic Editor 

PLOS ONE

Revision of the manuscript 

PONE-D-20-14264

“Intraoperative blood loss may be associated with myocardial injury after non-cardiac surgery”

Dear Editor,

Thank you for your letter dated June 24th, regarding our submitted manuscript. We appreciate the opportunity to resubmit our revised manuscript entitled “Intraoperative blood loss may be associated with myocardial injury after non-cardiac surgery”. We thank you for your constructive criticisms and suggestions for revision, which improved the presentation of our paper significantly. In this revision, we did our best to fully accommodate the comments and questions.

 The specific revisions and corrections made in point-by-point response to the editor and reviewers are presented in the following response letter. 

All authors have read and approved its submission to the PLOS ONE and have contributed significantly to this work. The whole manuscript or part of it, neither has been published and is not being considered for publication elsewhere in any language except as an abstract. None of the authors have any financial relationships with any company or any other bias or conflict of interest.

We hope that the revised version is now acceptable for publication in PLOS ONE.

.

Sincerely,

Seung-Hwa Lee, MD, Division of Cardiology, Department of Medicine, Heart Vascular Stroke Institute, Samsung Medical Center, Sungkyunkwan University School of Medicine, 81 Irwon-ro, Gangnam-gu, Seoul, Korea. 

Tel: +82-2-3410-3214; Fax: +82-2-3410-3849

E-mail address: shuaaa.lee@samsung.com

Response to the Reviewers

Reviewer #1.

Major comments

Introduction: our current understanding of MINS is not, as the authors state, principally myocardial oxygen supply/demand imbalance (which more accurately describes Type 2 myocardial infarction according to the 4th universal definition as published by the American College of Cardiology) but is pathophysiologically undefined. Exact causation has not been established but it is likely a heterogenous syndrome of cardiac myocyte necrosis secondary to ischemia, inflammation, immunological stress reaction (mostly through circulating cytokines such as TNF-alpha), and neuroendocrine dysregulation, and as such has a large number of potential clinical correlates. MINS is therefore more accurately a biomarker for increased postoperative morbidity and mortality from a range of sources.

>> Response: We thank you for your kind comments. Following your recommendation, we changed the Introduction section as below.

“During surgical procedures, massive blood loss can be frequently encountered and can lead to decreased hemoglobin level, and oxygen supply/demand mismatch has been proposed as one of mechanisms for the occurrence of MINS [3].” (Line 78)

The authors have excluded patients with conditions that cause raised troponin elevation (such as renal impairment, heart failure, sepsis, atrial fibrillation, pulmonary embolism etc.) as these are regarded as artefactual troponin rises. This is can be a legitimate decision, but requires justification. These conditions are often co-morbid with MINS as they share a common pathology (inflammatory response to tissue trauma), so true MINS incidence may be missed. This could be screened for using a sensitivity analysis, as was done for renal impairment and heart failure in the original VISION paper (1), where the MINS concept was first popularised. 

>> Response: We agree with reviewer that other conditions related to troponin elevation should have been more carefully analysed, so we added the sensitivity analysis as below.

“For sensitivity analysis, we evaluated whether the observed association was significant in patients with chronic kidney disease and heart failure and after January 2017 when the threshold of hemoglobin for intraoperative transfusion was lowered to 7 g/dL.” (Line 166)

“In the sensitivity analysis, this association was consistently found to be significant in patients with chronic kidney disease and heart failure and after lowering the threshold of hemoglobin for intraoperative transfusion in 2017 (S1 Table).” (Line 197)

“The observed association between intraoperative bleeding was shown to be significant in most of subgroups with risk factor for MINS.” (Line 287) 

Supplemental table 1. Sensitivity Analysis of the Observed Association between Significant Bleeding and Myocardial Injury after Noncardiac Surgery

　 OR (95% CI) P-value

Patients with heart failure (n=327) 4.63 (2.42-9.05) <0.001

Patients with chronic kidney disease (n=1012) 1.17 (0.86-1.59) <0.001

Surgeries after 2017 with the threshold of 

hemoglobin level < 7 g/dL for intraoperative transfusion 2.20 (1.94-2.50) <0.001

OR, odds ratio; CI, confidence interval; RBC, red blood cell 

There is a significant clinical and statistical difference in baseline characteristics between the bleeding and non-bleeding cohorts, with regard to anemia, alcohol use, chronic kidney disease, active cancer, being in ICU, continuous renal replacement therapy, ventilation, surgical risk, emergency operation status and requirement for continuous inotropic support. All of these affect the risk of MINS, and therefore confound the effect of bleeding on the incidence of MINS. This is a major limitation.

>> Response: We fully agree with the reviewer that there is a large difference in baseline characteristics of the two groups. So, we conducted inverse probability weighting for statistical adjustment for the difference. The result was not changed after mor rigrous statistical adjustment. Addistionally for clarification, we added in the limitation section that it could stll be biased by unmaseured confounders despite statistical adjustment.

” To retain a large sample size and maximize the study power while maintaining a balance in covariates between the two groups, we conducted rigorous adjustment for differences in baseline characteristics of patients using the weighted regression models with the inverse probability weighting (IPW) [19]. According to this technique, weights for patients without significant bleeding were the inverse of the propensity score and weights for patients with significant bleeding were the inverse of 1 – the propensity score.” (Line 160)

 “First, this is a single-center, observational study; the possibility of selection bias or unmeasured confounding factors exists. Also, our data may not be generalized to populations in other countries considering ethnic differences in blood management.” (Line 307) 

Table 1. Baseline characteristics according to significant intraoperative bleeding.

 Entire population IPW

　 No significant bleeding

(N = 13416) Significant bleeding

(N = 2510) ASD No significant bleeding

(N = 13370.5) Significant bleeding

(N = 2644.5) ASD

Male 8133 (60.6) 1562 (62.2) 3.3 8145.0 (60.9) 1658.9 (62.7) 3.7

Age 61.7 (±13.5) 60.0 (±13.5) 12.4 61.5 (±13.6) 62.2 (±13.3) 5.0

Preoperative anemia 4949 (36.9) 1646 (65.6) 59.9 5497.0 (41.1) 1064.8 (40.3) 1.7

Diabetes 7890 (58.8) 1748 (69.6) 22.7 8019.3 (60.0) 1446.4 (54.7) 10.7

Hypertension 7290 (54.3) 1424 (56.7) 4.8 7331.9 (54.8) 1436.0 (54.3) 1.1

Current smoking 1362 (10.2) 236 (9.4) 2.5 1337.6 (10.0) 282.8 (10.7) 2.3

Current alcohol 2798 (20.9) 360 (14.3) 17.2 2639.6 (19.7) 556.3 (21.0) 3.2

Chronic kidney disease 680 (5.1) 332 (13.2) 28.6 850.8 (6.4) 166.3 (6.3) 0.3

History of ischemic heart disease 1913 (14.3) 365 (14.5) 0.8 1939.1 (14.5) 395.5 (15.0) 1.3

History of heart failure 282 (2.1) 45 (1.8) 2.2 277.8 (2.1) 51.3 (1.9) 1.0

History of stroke 938 (7.0) 161 (6.4) 2.3 926.8 (6.9) 176.3 (6.7) 1.1

History of arrhythmia 882 (6.6) 161 (6.4) 0.6 875.4 (6.5) 179.4 (6.8) 0.9

History of heart valve disease 149 (1.1) 21 (0.8) 2.8 143.2 (1.1) 25.9 (1.0) 0.9

Active cancer 7422 (55.3) 1112 (44.3) 22.2 7216.2 (54.0) 1555.1 (58.8) 9.8

Preoperative care 

RBC transfusion 608 (4.5) 151 (6.0) 6.6 638.0 (4.8) 123.2 (4.7) 0.5

 Intensive care unit 424 (3.2) 278 (11.1) 31.2 574.7 (4.3) 111.4 (4.2) 0.4

 ECMO 0 1 (0.0) 2.8 0 0.2 (0.0) 1.1

 Continuous renal replacement therapy 18 (0.1) 54 (2.2) 19.1 54.7 (0.4) 11.3 (0.4) 0.3

 Ventilator 67 (0.5) 56 (2.2) 15.0 101.2 (0.8) 18.9 (0.7) 0.5

Operative variables 

 ESC/ESA surgical high risk 3997 (29.8) 1258 (50.1) 42.4 4306.0 (32.2) 748.8 (28.3) 8.5

 Emergency operation 1521 (11.3) 549 (21.9) 28.6 1745.6 (13.1) 354.4 (13.4) 1.0

 General anesthesia 13305 (99.2) 2500 (99.6) 5.5 13267.8 (99.2) 2610.5 (98.7) 5.2

 Operation duration, hours 3.94 (±2.14) 5.57 (±2.86) 64.4 4.20(±2.51 4.16±2.20) 1.7

 Continuous infusion of inotropes 4592 (34.2) 1278 (50.9) 34.2 4855.3 (36.3) 798.6 (30.2) 13.0

Types of surgery >0.99 

Vascular 981 (7.3) 169 (6.7) 

Orthopedic 667 (5.0) 146 (5.8) 

Neuro 3317 (24.7) 166 (6.6) 

Breast or Endo 148 (1.1) 22 (0.9) 

Plastic or Otolaryngeal or Eye 279 (2.1) 53 (2.1) 

Transplantation 300 (2.2) 768 (30.6) 

Gynecology or Urology 823 (6.1) 258 (10.3) 

Gastrointestinal 4559 (34.0) 741 (29.5) 

Noncardiac thoracic 2328 (17.4) 184 (7.3) 

Others 14 (0.1) 3 (0.1) 

Data are presented as n (%) or mean (±standard deviation)

IPW, inverse probability weighting; ASD, absolute standardized mean difference; RBC, red blood cell; ECMO, extracorporeal membranous oxygenation; RAAS, renin-angiotensin-aldosterone system; ESC, European Society of Cardiology; ESA, European Society of Anaesthesiology

The authors state that they have made an estimate of the effect of potential unmeasured confounders (presumably with a sensitivity analysis with a null variable although this is not explicitly stated). This technique requires more explanation, particularly in view of the very disparate and potentially powerfully confounding medical statuses of the two groups in the cohort. Sensitivity analysis is usually used in epidemiology to model the potential effect of a variable which is known or postulated from prior experience or research, but unmeasured in the study at hand. For example, “I think having X increases risk of MINS by OR 2, what effect would it have on my results if it 20% of my MINS group had X? What about 40%? Etc.” This enables you to make a statement such as “Even if X were present in one of my groups with prevalence up to 40%, it would not have changed the result”. How it was used in this study requires more explanation in the text. This technique does not compensate for the confounding caused by the major differences in medical status and operation type mentioned already.

>> Response: We agree that the statistical method and results of our sensitivity analysis needs more explanation. So, we added as below to the Method and Result sections.

“We also estimated the potential impact of unmeasured confounders by calculating the change of OR and CI according to the associations of unmeasured confounders with significant bleeding and MINS with an assumed unmeasured confounder with prevalence of 40% [20].” (Line 168)

“Significant bleeding was associated with MINS, even if the assumed unmeasured confounders were present in one of our groups with prevalence up to 40%. (S2 Table)” (Line 199)

The authors analyse their data with two different definitions of significant blood loss (including and excluding whether transfusion was required). This is confusing. They could consider picking one definition of bleeding and analysing that as the primary outcome, and then if they wanted to explore the interaction of transfusion they could do that as a secondary analysis (perhaps using matched pairs controlled for hemoglobin drop or some similar technique). Alternatively, they could divide the groups into a two-by-two table (hemoglobin decrease/transfusion) and just make this the primary reported result (as they did in table s5), skipping the intermediate steps. This would be an important result, as transfusion has it’s own risks that may be associated with MINS, and disentangling it from the effect of the bleeding event itself, and anemia, is important.

>> Response: We agree that presenting two analyses might be confusing for ther readers. Following your recommendation, we only used definition of significant blood loss as our primary analysis. We conducted for group comparison according to bleeding and intraoperative transfusion and the analysis according to the actual decrease of hemoglobin was presented as a supplemental table. 

“The patients for this study were divided according to significant intraoperative bleeding. Significant intraoperative bleeding was defined as an absolute value of intraoperative lowest hemoglobin level < 7 g/dL, relative value intraoperative lowest hemoglobin level less than 50% of preoperative measurement, or need for intraoperative packed RBC transfusion [10-13].” (Line 116)

“After an adjustment with multivariable analysis, the risk for MINS was found to be significantly increased for the significant bleeding group (35.2% vs. 16.4%; OR, 1.80; 95% CI, 1.61–2.00; p < 0.001) (Table 2). The result after IPW adjustment also showed an increased risk for MINS for the significant bleeding group (OR, 1.58; 95% CI, 1.43–1.75; p < 0.001) (Table 2).” (Line 189)

“Compared with the no hemoglobin decrease without transfusion group, the risk for MINS increased according to both hemoglobin decrease and receipt of RBC transfusion (OR, 2.04; 95% CI, 1.83–2.27; p < 0.001 for no hemoglobin decrease with transfusion; OR, 6.13; 95% CI, 4.81–7.82; p < 0.001 for hemoglobin decrease without transfusion; and OR, 8.66; 95% CI, 6.77–10.96; p < 0.001 for hemoglobin decrease with transfusion) (Table 3). When the patients solely divided according to hemoglobin decrease without considering RBC transfusion, 15,353 (96.4%) patients were in the no hemoglobin decrease group and 573 (3.6%) patients were in the hemoglobin decrease group (S4 Table), the incidence of MINS was substantially increased in the hemoglobin decrease group (58.3% vs. 17.9%; OR, 3.28; 95% CI, 2.70–4.00; p < 0.001) (S5 and S6 Tables).” (Line 209) 

Table 2. The incidence of myocardial injury after noncardiac surgery and mortality.

 Univariable analysis Multivariable analysis IPW analysis

　

　 No significant bleeding

(N = 13416) Significant bleeding

(N = 2510) Unadjusted OR/HR (95% CI) p value Adjusted OR/HR (95% CI) p value Adjusted OR/HR (95% CI) p value

MINS 2200 (16.4) 883 (35.2) 2.77 (2.52-3.04) < 0.001 1.80 (1.61-2.00) < 0.001 1.58 (1.43-1.75) < 0.001

30-day mortality 173 (1.3) 112 (4.5) 3.52 (2.77-4.46) < 0.001 2.04 (1.55-2.67) < 0.001 2.51 (1.91-3.28) < 0.001

 Cardiovascular 50 (0.4) 28 (1.1) 3.04 (1.91-4.82) < 0.001 1.92 (1.13-3.27) 0.02 1.90 (1.10-3.29) < 0.001

 Noncardiovascular 123 (0.9) 84 (3.3) 3.71 (2.81-4.90) < 0.001 2.10 (1.53-2.88) < 0.001 2.76 (2.02-3.76) < 0.001

Data are presented as n (%)

MINS was presented with OR, and mortalities were presented as HR

IPW, inverse probability weighting; MINS, myocardial injury after noncardiac surgery; OR, odds ratio; HR, hazard ratio; CI, confidence interval 

Table 3. The incidence of myocardial after noncardiac surgery and mortality according to intraoperative hemoglobin decrease and transfusion.

　 No hemoglobin decrease without transfusion

(N = 13416) No hemoglobin decrease with transfusion

(N = 1937) Hemoglobin decrease without transfusion

(N = 275) Hemoglobin decrease with transfusion

(N = 298)

　 

MINS, No (%) 2200 (16.4) 549 (28.3) 147 (53.5) 187 (62.8)

 Unadjusted OR (95% CI) 1 [reference] 2.02 (1.81-2.24) 5.85 (4.60-7.46) 8.59 (6.77-10.94)

 p value < 0.001 < 0.001 < 0.001

30-day mortality, No (%) 173 (1.3) 61 (3.1) 26 (9.5) 25 (8.4)

 Unadjusted HR (95% CI) 2.46 (1.84-3.30) 7.75 (5.14-11.71) 6.76 (4.45-10.29)

 p value < 0.001 < 0.001 < 0.001

Cardiovascular mortality, No (%) 50 (0.4) 17 (0.9) 9 (3.3) 2 (0.7)

 Unadjusted HR (95% CI) 2.37 (1.37-4.11) 9.23 (4.54-18.78) 1.87 (0.45-7.67)

 p value < 0.001 < 0.001 0.39

Non-cardiovascular mortality, No (%) 123 (0.9) 44 (2.3) 17 (6.2) 23 (7.7)

 Unadjusted HR (95% CI) 2.50 (1.77-3.52) 7.14 (4.30-11.86) 8.76 (5.61-13.67)

 p value < 0.001 < 0.001 < 0.001

Data are presented as n (%)

MINS was presented with OR, and mortalities were presented as HR

MINS, myocardial injury after noncardiac surgery; OR, odds ratio; HR, hazard ratio; CI, confidence interval 

Supplemental table 4. Baseline Characteristics According to the Actual Hemoglobin Decrease without Regarding Intraoperative Transfusion

　 No hemoglobin decrease Hemoglobin decrease P Value

 (N = 15353) (N = 573) 

Male 9387 (61.1) 308 (53.8) <0.001

Age 61.6 (±13.5) 55.8 (±13.5) <0.001

Preoperative anemia 6119 (39.9) 476 (83.1) <0.001

Diabetes 9219 (60.0) 419 (73.1) <0.001

Hypertension 8401 (54.7) 313 (54.6) 0.99

Current smoking 1527 (9.9) 71 (12.1) 0.07

Current alcohol 3063 (20.0) 95 (16.6) 0.05

Chronic kidney disease 907 (5.9) 105 (18.3) <0.001

History of ischemic heart disease 2212 (14.4) 66 (11.5) 0.06

History of heart failure 322 (2.1) 5 (0.9) 0.06

History of stroke 1062 (6.9) 37 (6.5) 0.73

History of arrhythmia 1021 (6.7) 22 (3.8) 0.01

History of heart valve disease 165 (1.1) 5 (0.9) 0.8

Active cancer 8372 (54.5) 162 (28.3) <0.001

Preoperative care 

 RBC transfusion 695 (4.5) 64 (11.2) <0.001

 Intensive care unit 570 (3.7) 132 (23.0) <0.001

 ECMO 1 (0.0) 0 >0.99

 Continuous renal replacement therapy 37 (0.2) 35 (6.1) <0.001

 Ventilator 90 (0.6) 33 (5.8) <0.001

Operative variables 

 ESC/ESA surgical high risk 4905 (31.9) 350 (61.1) <0.001

 Emergency operation 1854 (12.1) 216 (37.7) <0.001

 General anesthesia 15235 (99.2) 570 (99.5) 0.68

 Operation duration, hours 4.09 (±2.24) 6.95 (±3.26) <0.001

 Continuous infusion of inotropes 5483 (35.7) 387 (67.5) <0.001

 Types of surgery 

Vascular 1090 (7.1) 31 (5.4) 

Orthopedic 800 (5.2) 28 (4.9) 

Neuro 3449 (22.5) 36 (6.3) 

Breast or Endo 165 (1.1) 5 (0.9) 

Plastic or Otolaryngeal or Eye 318 (2.1) 15 (2.6) 

Transplantation 773 (5.0) 295 (51.5) 

Gynecology or Urology 1062 (6.9) 38 (6.6) 

Gastrointestinal 5176 (33.7) 112 (19.5) 

Noncardiac thoracic 2494 (16.2) 13 (2.3) 

Others 26 (0.2) 0 

Data are presented as n (%) or mean (±standard deviation)

RBC, red blood cell; ECMO, extracorporeal membranous oxygenation; RAAS, renin-angiotensin-aldosterone system; ESC, European Society of Cardiology; ESA, European Society of Anaesthesiology 

Supplemental table 5. The Incidence of Myocardial Injury after Noncardiac Surgery and Mortality According to the Actual Hemoglobin Decrease without Regarding Intraoperative Transfusion

　

　 No hemoglobin decrease

(N = 15353) Hemoglobin decrease

(N = 573) Unadjusted OR/HR (95% CI) p value Adjusted OR/HR (95% CI) p value

MINS 2749 (17.9) 334 (58.3) 6.41 (5.40-7.61) < 0.001 3.28 (2.70-4.00) < 0.001

30-day mortality 234 (1.5) 51 (8.9) 6.11 (4.52-8.27) < 0.001 2.52 (1.75-3.63) < 0.001

 Cardiovascular 67 (0.4) 11 (1.9) 4.58 (2.42-8.66) < 0.001 2.01 (0.94-4.32) 0.07

 Noncardiovascular 167 (1.1) 40 (7.0) 6.73 (4.76-9.50) < 0.001 2.73 (1.80-4.13) < 0.001

Minor comments

Line 135 – I assume “high risk” here refers to operative mortality risk >5% in table 3 of the ESC/ESA guidelines 2014 (2)? This could be made absolutely clear.

>> Response: Following your recommendation, we clarified the definition.

“High-risk surgery was defined as procedures with mortality risk >5% according to the 2014 European Society of Cardiology/Anesthesiology guidelines [17].” (Line 131)

Line 65 and elsewhere – the authors give their main result as “16.4% vs. 35.2%; odds ratio, 1.80; 95% confidence interval, 1.61–2.00; p < 0.001” This should read “35.2% vs 16.4%; odds ratio 1.80; 95% confidence interval 1.61-2.00; p<0.001”. This is true for all the odds ratios in the text.

>> Response: Following your recommendation, we changed the orders throughout the Abstract and Results sections.

“After an adjustment with inverse probability weighting, the incidence of MINS was higher in the significant bleeding group (35.2% vs. 16.4%; odds ratio, 1.58; 95% confidence interval, 1.43–1.75; p < 0.001).” (Line 65)

“After an adjustment with multivariable analysis, the risk for MINS was found to be significantly increased for the significant bleeding group (35.2% vs. 16.4%; OR, 1.80; 95% CI, 1.61–2.00; p < 0.001) (Table 2).” (Line 189)

“When the patients solely divided according to hemoglobin decrease without considering RBC transfusion, 15,353 (96.4%) patients were in the no hemoglobin decrease group and 573 (3.6%) patients were in the hemoglobin decrease group (S4 Table), the incidence of MINS was substantially increased in the hemoglobin decrease group (58.3% vs. 17.9%; OR, 3.28; 95% CI, 2.70–4.00; p < 0.001) (S5 and S6 Tables).” (Line 214)

While the authors tell us that these are patients having high risk surgery, they do not include a breakdown of the types of surgery performed in the table 1 (Baseline Characteristics). This is important enough to be in the main text, as it affects the applicability of these findings to our individual practice settings. In addition, some surgical populations (vascular surgery in particular) are at very high risk of MINS, and a different incidence between groups may confound the data. This should be moved up into the main text from the supplement (from table S1 into table 1). P values should also be provided.

>> Response: We agree that this needs to be included in the main table. Following your recommendation, we included the types of surgery in the Table 1. However, including the types of surgery as an adjustment variable would be a double adjustment with the risk of surgery. Moreover, it is true that the risk of MINS is reported to be higher for some types of surgery, but it is also likely that the surgeries with different risks are classified within the same type of surgery. For clarification, we also added that our results could be different according to the types of surgery.

“In In addition, this study contains all types of noncaridac surgery, and the result might be different in particular types of surgery.” (Line 310)

Line 130 – the exclusion criteria of factors that artefactually increase MINS should be moved up into the methods section for clarity.

>> Response: Following your recommendation, we moved this section to the Method section.

“Among 3,193 patients with postoperative cTn elevation, 110 patients were diagnosed with non-ischemic cause, and 3083 (19.4%) patients were diagnosed with MINS.” (Line 126)

General – this dataset spans a relatively long period of time, beginning in 2010, during which our understanding of the hazards of transfusion has evolved and trigger levels of hemoglobin for transfusion have dropped from 10 to 7 g/dl. Were there changes to the author’s institutional policy or consensus in practice on transfusion triggers during this time? Did the authors consider a temporal analysis either on lowest hemoglobin before transfusion, or numbers of transfusions?

>> Response: We agree that including the change of the institutional protocol into an analysis would be helpful for readers. So, we added as below to the Method, Result, and Discussion sections as a sensitivity analysis. 

““For sensitivity analysis, we evaluated whether the observed association was significant in patients with chronic kidney disease and heart failure and after January 2017 when the threshold of hemoglobin for intraoperative transfusion was lowered to 7 g/dL.” (Line 166)

“In the sensitivity analysis, this association was consistently found to be significant in patients with chronic kidney disease and heart failure and after lowering the threshold of hemoglobin for intraoperative transfusion in 2017 (S1 Table).” (Line 197)

“The observed association between intraoperative bleeding was shown to be significant in most of subgroups with risk factor for MINS.” (Line 287) 

Supplemental table 1. Sensitivity Analysis of the Observed Association between Significant Bleeding and Myocardial Injury after Noncardiac Surgery

　 OR (95% CI) P-value

Patients with heart failure (n=327) 4.63 (2.42-9.05) <0.001

Patients with chronic kidney disease (n=1012) 1.17 (0.86-1.59) <0.001

Surgeries after 2017 with the threshold of 

hemoglobin level < 7 g/dL for intraoperative transfusion 2.20 (1.94-2.50) <0.001

OR, odds ratio; CI, confidence interval; RBC, red blood cell

Details of the power analysis outside of prospective studies are not required, even though they can be calculated, p values are sufficient. Medical readers without a research or statistics background may be confused. 

>> Response: Following your recommendation, we removed the power analysis.

Reviewer #2.

Database/registry study with many limitations, most of which were addressed in the text of the paper. They acknowledged that more studies (more rigorously conducted studies to be specific) need to be conducted to support their conclusions and essentially this study is mostly hypothesis generating.

 Large population included in study and overall well conducted (given limitations of registry) so does add some controversy to literature that is worth considering (changing transfusion 'triggers' in high risk populations to avoid MINS).

>> Response: We thank you for your kind comments.

 There were significant differences in baseline characteristics in patients who had blood loss versus those that didn't therefore there were many confounding factors that could influence reasons for these patients to have more MINS besides just blood loss, not sure all of this was accounted for by the statistics or fully explained/justified.

>> Response: We fully agree with the reviewer that there is a large difference in baseline characteristics of the two groups. So, we conducted inverse probability weighting for statistical adjustment for the difference. The result was not changed after mor rigrous statistical adjustment.

” To retain a large sample size and maximize the study power while maintaining a balance in covariates between the two groups, we conducted rigorous adjustment for differences in baseline characteristics of patients using the weighted regression models with the inverse probability weighting (IPW) [19]. According to this technique, weights for patients without significant bleeding were the inverse of the propensity score and weights for patients with significant bleeding were the inverse of 1 – the propensity score.” (Line 160)

For clarification, we added in the limitation section that it could stll be biased by unmaseured confounders despite statistical adjustment.

“First, this is a single-center, observational study; the possibility of selection bias or unmeasured confounding factors exists. Also, our data may not be generalized to populations in other countries considering ethnic differences in blood management.” (Line 307) 

Table 1. Baseline characteristics according to significant intraoperative bleeding.

 Entire population IPW

　 No significant bleeding

(N = 13416) Significant bleeding

(N = 2510) ASD No significant bleeding

(N = 13370.5) Significant bleeding

(N = 2644.5) ASD

Male 8133 (60.6) 1562 (62.2) 3.3 8145.0 (60.9) 1658.9 (62.7) 3.7

Age 61.7 (±13.5) 60.0 (±13.5) 12.4 61.5 (±13.6) 62.2 (±13.3) 5.0

Preoperative anemia 4949 (36.9) 1646 (65.6) 59.9 5497.0 (41.1) 1064.8 (40.3) 1.7

Diabetes 7890 (58.8) 1748 (69.6) 22.7 8019.3 (60.0) 1446.4 (54.7) 10.7

Hypertension 7290 (54.3) 1424 (56.7) 4.8 7331.9 (54.8) 1436.0 (54.3) 1.1

Current smoking 1362 (10.2) 236 (9.4) 2.5 1337.6 (10.0) 282.8 (10.7) 2.3

Current alcohol 2798 (20.9) 360 (14.3) 17.2 2639.6 (19.7) 556.3 (21.0) 3.2

Chronic kidney disease 680 (5.1) 332 (13.2) 28.6 850.8 (6.4) 166.3 (6.3) 0.3

History of ischemic heart disease 1913 (14.3) 365 (14.5) 0.8 1939.1 (14.5) 395.5 (15.0) 1.3

History of heart failure 282 (2.1) 45 (1.8) 2.2 277.8 (2.1) 51.3 (1.9) 1.0

History of stroke 938 (7.0) 161 (6.4) 2.3 926.8 (6.9) 176.3 (6.7) 1.1

History of arrhythmia 882 (6.6) 161 (6.4) 0.6 875.4 (6.5) 179.4 (6.8) 0.9

History of heart valve disease 149 (1.1) 21 (0.8) 2.8 143.2 (1.1) 25.9 (1.0) 0.9

Active cancer 7422 (55.3) 1112 (44.3) 22.2 7216.2 (54.0) 1555.1 (58.8) 9.8

Preoperative care 

RBC transfusion 608 (4.5) 151 (6.0) 6.6 638.0 (4.8) 123.2 (4.7) 0.5

 Intensive care unit 424 (3.2) 278 (11.1) 31.2 574.7 (4.3) 111.4 (4.2) 0.4

 ECMO 0 1 (0.0) 2.8 0 0.2 (0.0) 1.1

 Continuous renal replacement therapy 18 (0.1) 54 (2.2) 19.1 54.7 (0.4) 11.3 (0.4) 0.3

 Ventilator 67 (0.5) 56 (2.2) 15.0 101.2 (0.8) 18.9 (0.7) 0.5

Operative variables 

 ESC/ESA surgical high risk 3997 (29.8) 1258 (50.1) 42.4 4306.0 (32.2) 748.8 (28.3) 8.5

 Emergency operation 1521 (11.3) 549 (21.9) 28.6 1745.6 (13.1) 354.4 (13.4) 1.0

 General anesthesia 13305 (99.2) 2500 (99.6) 5.5 13267.8 (99.2) 2610.5 (98.7) 5.2

 Operation duration, hours 3.94 (±2.14) 5.57 (±2.86) 64.4 4.20(±2.51 4.16±2.20) 1.7

 Continuous infusion of inotropes 4592 (34.2) 1278 (50.9) 34.2 4855.3 (36.3) 798.6 (30.2) 13.0

Types of surgery >0.99 

Vascular 981 (7.3) 169 (6.7) 

Orthopedic 667 (5.0) 146 (5.8) 

Neuro 3317 (24.7) 166 (6.6) 

Breast or Endo 148 (1.1) 22 (0.9) 

Plastic or Otolaryngeal or Eye 279 (2.1) 53 (2.1) 

Transplantation 300 (2.2) 768 (30.6) 

Gynecology or Urology 823 (6.1) 258 (10.3) 

Gastrointestinal 4559 (34.0) 741 (29.5) 

Noncardiac thoracic 2328 (17.4) 184 (7.3) 

Others 14 (0.1) 3 (0.1) 

Data are presented as n (%) or mean (±standard deviation)

IPW, inverse probability weighting; ASD, absolute standardized mean difference; RBC, red blood cell; ECMO, extracorporeal membranous oxygenation; RAAS, renin-angiotensin-aldosterone system; ESC, European Society of Cardiology; ESA, European Society of Anaesthesiology

Few errors in supplemental tables that need to be corrected - Supplemental Table 1 has a category of surgery "Orthopediatric" since all patients were above age 18 I can only assume this should be 'Orthopedic'. Supplemental Table 3 says "ionotropics" should read 'ionotropes'

>> Response: These points were clearly our mistake. We changed “Orthopediatric” to “Orthopedic”, and “inotropics” to “inotropes”

---

## [Decision Letter · Decision Letter 1]

11 Aug 2020

PONE-D-20-14264R1

Intraoperative blood loss may be associated with myocardial injury after non-cardiac surgery

PLOS ONE

Dear Dr. Lee,

Thank you for submitting your manuscript to PLOS ONE. After careful consideration, we feel that it has merit but does not fully meet PLOS ONE’s publication criteria as it currently stands. Therefore, we invite you to submit a revised version of the manuscript that addresses the points raised during the review process.

Would like the authors to:

A. Please address the comments from reviewer 2 and review the manuscript in its entirety to avoid further grammar mistakes.

B. Please clarify in your response letter the potential overlap between this manuscript and the one submitted to the European Journal of Anaesthesiology on a similar topic. Please delineate the items that are similar and the ones which are unique to this contribution to avoid duplication.

We look forward to receiving your revised manuscript.

Kind regards,

Wen-Chih Wu, MD, MPH

Academic Editor

PLOS ONE

1. It has come to our attention that a similar manuscript from your group "Association between preoperative anemia and myocardial injury after noncardiac surgery" by Dr Seung-Hwa Lee is at revision at the European Journal of Anaesthesiology. In oder to assess the distinction between these manuscripts, please provide us with a copy of the current version of your manuscript under review at the European Journal of Anaesthesiology and provide us with a discussion of how this work differs from the work in your current manuscript at PLOS ONE. This information will be used for internal purposes only, and will not be published if your paper is accepted at our journal. Please also let us know if your manuscript is to be published at the European Journal of Anaesthesiology, in which case you would need to include a discussion in this manuscript of how the works differ. Please feel free to email shepp@plos.org with any questions.

Reviewers' comments:

Reviewer's Responses to Questions

2. Is the manuscript technically sound, and do the data support the conclusions?

Reviewer #2: Partly

3. Has the statistical analysis been performed appropriately and rigorously? 

Reviewer #2: I Don't Know

4. Have the authors made all data underlying the findings in their manuscript fully available?

Reviewer #2: Yes

5. Is the manuscript presented in an intelligible fashion and written in standard English?

Reviewer #2: Yes

6. Review Comments to the Author

Reviewer #2: Many areas have been clarified and statistical analysis has improved from previous submission.

There are still some grammatical errors and confusion in the revised submission.

Line 55 - should read: which is 'a' severe and common postoperative complication.

Line 79 - should read: one of 'the' mechanisms

Line 193 - does not make sense grammatically (the word interacted), do you mean "the subgroup analysis demonstrated....was 'confounded' by emergency surgery?

Line 259 - additional reference (although done in cardiac surgery, would still support your statements) Transfusion, Jan 10, 2008. The influence of baseline hemoglobin on tolerance of anemia in cardiac surgery. Karkouti et. al.

Line 293 - should read: When the patients 'were' solely divided....

Line 284 - Although your conclusions are more or less consistent with the idea of restrictive transfusion, they do not completely support the current transfusion thresholds, your results from this study would suggest that the lower trigger limit should perhaps be higher than 7g/dL (although you do mention this point later)

Line 288 - the explanation for lack of correlation of emergency surgery with MINS makes no sense, one would expect that unstable bleeding patients would experience a greater propensity for MINS (since more supply/demand mismatch due to increased work on the heart and increased cardiac output in these situations with an increase in cytokines/inflammatory markers)

7. PLOS authors have the option to publish the peer review history of their article (what does this mean?). If published, this will include your full peer review and any attached files.

Reviewer #2: No

---

## [Author Response · Author response to Decision Letter 1]

13 Sep 2020

Response to the Editor

A. Please address the comments from reviewer 2 and review the manuscript in its entirety to avoid further grammar mistakes.

>> Response: We thank you for your kind comments. Following your recommendation, we had the whole script proofread by a professional editor.

B. Please clarify in your response letter the potential overlap between this manuscript and the one submitted to the European Journal of Anaesthesiology on a similar topic. Please delineate the items that are similar and the ones which are unique to this contribution to avoid duplication.

>> Response: Regarding this, the reviwer and we have exchanged the letters. We explained to the reviewer as below.

“These two studies share the entire cohort, but the enrolled study populations of the two studies differ as well as their conclusions and the clinical implications. 

Using the entire cohort, we first analzed the effect of "preoperative anemia' after enrolling all patients with preoeprative hemoglobin level. However, we also felt the need for further analysis on the effect of intraoperative hemoglobin level, because the recent concept of patient blood management covers maintaining an adequate hemoglobin level during pre-, intra, and postoperative periods. Owing to the retrospective nature of our cohort, not every patients with preoperative hemoglobin level had an intraoperartive measurement. So, for intraoperative hemoglobin level, we had to conduct a separate analysis after exculding those without an intraoperative measurement. The study on "preoperative anemia" was conducted among 35,170 patients, and this study on "intraoperative hemoglobin level" exluded 19,2297 patients without intraoperative hemoglobin level and was conducted in 15,926 patients.

These two studies highlight the importance on maintaining an adequate hemoglobin level to decrease MINS, but during the different period by using different study populations.

 I would like to clarify that we had no intention to duplicate or hide our former analysis, but it was an inevitable decision for us to conduct separate analyses for pre- and intraopearive hemoglobin level.”

Now that the reviwer understands that those two studies are distinct. We also mentioned the result of our analysis on preoperative anemia as below.

“Despite various possible causes, the most frequently proposed mechanism for MINS is oxygen supply/demand mismatch, and our previous study also showed that preoperative anemia was associated with MINS [21]. In this follow-up study, we demonstrated that blood loss during surgical procedures could also increase MINS. So, our results on the association between intraoperative hemoglobin decrease and the increased incidence of MINS can be explained by reduced oxygen-carrying capacity compromising the myocardial oxygen supply while simultaneously requiring higher cardiac output to maintain adequate systemic circulation [22].” 

Response to the Reviewers

Reviewer #1.

1. It has come to our attention that a similar manuscript from your group "Association between preoperative anemia and myocardial injury after noncardiac surgery" by Dr Seung-Hwa Lee is at revision at the European Journal of Anaesthesiology. In oder to assess the distinction between these manuscripts, please provide us with a copy of the current version of your manuscript under review at the European Journal of Anaesthesiology and provide us with a discussion of how this work differs from the work in your current manuscript at PLOS ONE. This information will be used for internal purposes only, and will not be published if your paper is accepted at our journal. Please also let us know if your manuscript is to be published at the European Journal of Anaesthesiology, in which case you would need to include a discussion in this manuscript of how the works differ. Please feel free to email shepp@plos.org with any questions. 

>> Response: We thank you for your kind comments. As we dicussed on e-mails, the results of the analysis on preoperative anemia is mentioned in the script as below with a citation that needs to be confirmed after the publication of the former study.

“Despite various possible causes, the most frequently proposed mechanism for MINS is oxygen supply/demand mismatch, and our previous study also showed that preoperative anemia was associated with MINS [21]. In this follow-up study, we demonstrated that blood loss during surgical procedures could also increase MINS. So, our results on the association between intraoperative hemoglobin decrease and the increased incidence of MINS can be explained by reduced oxygen-carrying capacity compromising the myocardial oxygen supply while simultaneously requiring higher cardiac output to maintain adequate systemic circulation [22].” 

When our previous work is ready to be cited , we will add it as reference 21. 

Reviewer #2.

Line 55 - should read: which is 'a' severe and common postoperative complication.

>> Response: We thank you for your kind comments. We changed all the montioned parts as below and had the whole script proofread by professional editor.

“This study aimed to evaluate the association between intraoperative blood loss and myocardial injury after non-cardiac surgery (MINS), which is a severe and common postoperative complication.”

Line 79 - should read: one of 'the' mechanisms

>> Response: We changed it as below.

“Massive blood loss is frequently encountered during surgical procedures and can lead to decreased hemoglobin levels. Oxygen supply/demand mismatch is a proposed as one of the mechanism for MINS [3].”

Line 193 - does not make sense grammatically (the word interacted), do you mean "the subgroup analysis demonstrated....was 'confounded' by emergency surgery?

>> Response: We changed it as below.

“Subgroup analysis demonstrated that the association between significant bleeding and MINS was confounded by emergency surgery.”

Line 259 - additional reference (although done in cardiac surgery, would still support your statements) Transfusion, Jan 10, 2008. The influence of baseline hemoglobin on tolerance of anemia in cardiac surgery. Karkouti et. al.

>> Response: Following your recommendation, we added the reference.

“In this study, we selected patients whose intraoperative hemoglobin level decreased below 7 g/dL, which is generally assumed to be the level that most patients can tolerate [10-12,23].”

Line 213 - should read: When the patients 'were' solely divided....

>> Response: We changed it as below.

“When the patients were solely divided according to hemoglobin decrease without considering RBC transfusion, 15,353 (96.4%) patients had no hemoglobin decrease whereas 573 (3.6%) patients did have decreased hemoglobin (S4 Table). The incidence of MINS was substantially increased in the decreased hemoglobin group (58.3% vs. 17.9%; OR, 3.28; 95% CI, 2.70–4.00; p < 0.001) (S5 and S6 Tables).”

Line 284 - Although your conclusions are more or less consistent with the idea of restrictive transfusion, they do not completely support the current transfusion thresholds, your results from this study would suggest that the lower trigger limit should perhaps be higher than 7g/dL (although you do mention this point later)

>> Response: Following your recommendation, we changed it as below.

“Despite the current guidelines advocating restrictive use of blood transfusion during surgical procedures, our findings suggest that the hemoglobin threshold for intraoperative transfusion may be higher than 7g/dL in order to prevent MINS [10-12,17].”

Line 288 - the explanation for lack of correlation of emergency surgery with MINS makes no sense, one would expect that unstable bleeding patients would experience a greater propensity for MINS (since more supply/demand mismatch due to increased work on the heart and increased cardiac output in these situations with an increase in cytokines/inflammatory markers) 

>> Response: This sentence needed more explanation. We agree that unstable bleeding events act as a cause of MINS. What we meant was that these preoperative events might have already caused enough metabolic stress for MINS and diluted the direct effect of intraoperative blood loss. For clarification, we added as below to the script.

“This may be because patients requiring emergency surgery include those with preoperative hemodynamic instability or massive bleeding, which could be strongly associated with cTn elevation. These events causing a large fluctuation of hemoglobin levels from the preoperative period might have diluted the effect of intraoperative blood loss.”

---

## [Decision Letter · Decision Letter 2]

9 Oct 2020

Intraoperative blood loss may be associated with myocardial injury after non-cardiac surgery

PONE-D-20-14264R2

Dear Dr. Lee,

We’re pleased to inform you that your manuscript has been judged scientifically suitable for publication and will be formally accepted for publication once it meets all outstanding technical requirements.

Kind regards,

Wen-Chih Hank Wu, MD, MPH

Academic Editor

PLOS ONE

Additional Editor Comments (optional):

Reviewers' comments:

Reviewer's Responses to Questions

**Comments to the Author**

1. If the authors have adequately addressed your comments raised in a previous round of review and you feel that this manuscript is now acceptable for publication, you may indicate that here to bypass the “Comments to the Author” section, enter your conflict of interest statement in the “Confidential to Editor” section, and submit your "Accept" recommendation.

Reviewer #2: All comments have been addressed

2. Is the manuscript technically sound, and do the data support the conclusions?

Reviewer #2: Partly

3. Has the statistical analysis been performed appropriately and rigorously? 

Reviewer #2: I Don't Know

4. Have the authors made all data underlying the findings in their manuscript fully available?

Reviewer #2: Yes

5. Is the manuscript presented in an intelligible fashion and written in standard English?

Reviewer #2: Yes

6. Review Comments to the Author

Reviewer #2: Have not referenced previous work that is based on the same cohort of patients, perhaps it is not yet published? Still have one grammatical error on line 79, "Oxygen supply/demand mismatch is a proposed as one of the mechanism for MINS [3]" remove "a"

7. PLOS authors have the option to publish the peer review history of their article (what does this mean?). If published, this will include your full peer review and any attached files.

Reviewer #2: No

---

## [Editor Report · Acceptance letter]

8 Feb 2021

PONE-D-20-14264R2 

Intraoperative blood loss may be associated with myocardial injury after non-cardiac surgery 

Dear Dr. Lee:

I'm pleased to inform you that your manuscript has been deemed suitable for publication in PLOS ONE. Congratulations! Your manuscript is now with our production department. 

Kind regards, 

on behalf of

Dr. Wen-Chih Hank Wu 

Academic Editor

PLOS ONE